# Urban Flora Structure and Carbon Storage Potential of Woody Trees in Different Land Use Units of Cotonou (West Africa)

**Assouhan Jonas Atchadé** [1,2,*] , **Madjouma Kanda** [2], **Fousseni Folega** [2] , **Abdoul Aziz Diouf** [3],
**Symphorien Agbahoungba** [4] , **Marra Dourma** [2], **Kperkouma Wala** [2] **and Koffi Akpagana** [2]

[1] Regional Center of Excellence on Sustainable Cities in Africa (CERViDA-DOUNEDON) of the University of Lomé, Lomé 01 BP 1515, Togo
[2] Laboratory of Botany and Plant Ecology (LBEV), Faculty of Science, University of Lomé, Lomé 01 BP 1515, Togo
[3] Center de Suivi Ecologique, Fann Résidence, Dakar BP 15532, Senegal
[4] Laboratory of Applied Ecology, Faculty of Agronomic Sciences, University of Abomey-Calavi, Cotonou 01 BP 526, Benin
[*] Correspondence: assouhan.atchade@cervida-togo.org

**Abstract:** Urbanization is a current concern, particularly in Africa, where it is expected to continue and increasingly threaten the effectiveness of plant biodiversity, natural carbon sinks, and the sustainability of cities. This paper investigates the structural parameters and carbon storage potential of trees in the land use units of the city of Cotonou in southern Benin. A total of 149 plots at 2500 m$^2$ each were randomly generated, and trees with a diameter $\geq$ 10 cm were inventoried. ANOVA revealed that the means of structural parameters (diameter and height classes) and carbon stock rate varied significantly ($p < 0.001$) across land use units in the city. Tree basal area is estimated at 4.52 $\pm$ 5.24 m$^2$ ha$^{-1}$, with an average of 12.72 (13) feet ha$^{-1}$. The average diameter of the trees is estimated at 57.94 $\pm$ 29.71 cm. Approximately 1000 kg ha$^{-1}$ (0.94 $\times$ 10$^3$ kg ha$^{-1}$) of carbon is stored in the city of Cotonou. Green spaces (1.21 $\times$ 10$^3$ kg ha$^{-1}$) and roads (1.19 $\times$ 10$^3$ kg ha$^{-1}$) are the units that recorded the highest carbon stocks. *Khaya senegalensis*, *Mangifera indica*, and *Terminalia mentally* lead the top ten species with high stock potential. This study demonstrates the contribution of urban trees to global atmospheric carbon reduction, which varies by species, land use units, and tree density. Future research could investigate an i-Tree Landscape approach for urban carbon estimation. This could reinforce urban carbon data availability for urban ecological planning.

**Keywords:** urban trees; carbon storage; ecological infrastructure; climate change; mitigation; urban carbon footprint; urban greening; Cotonou

## 1. Introduction

Cities are spreading exponentially, home to about 55% of the world's population today, and this figure is expected to reach 60% by 2030, while 90% of this increase occurs in Asia and Africa [1]. In Africa, the urban population is expected to continue to grow from 36% in 2010 to 50% in 2030 [2]. African cities are particularly affected with spontaneous, uncontrolled, and environmentally damaging urbanization [3]. These disturbing changes do not exclude Benin, a West African country, where the urbanization rate increased from 11% in 1960 to 40% in 1990, then from 42% in 2005 to 44% in 2015 and will reach 56.2% in 2025 [4].

Urban expansion is a major cause of climate change through land use change, biodiversity loss, invasion, and pollution [5]. These environmental challenges also threaten the sustainability of the world's cities, which to a large extent depends on the nature and extent of ecosystem services provided by the biodiversity in urban areas [6,7]. Because many urban areas are ecologically sensitive, the prevalence of urban areas with large human populations will have a negative effect on urban green space, urban health, urban



climate, urban environment, urban planning [8,9], and the socioeconomic well-being of urban residents [10]. Urban areas are the largest source of greenhouse gas emissions, as 75% of carbon emissions come from cities and are responsible for 70–80% of global energy consumption [5]. However, urbanization does not lead to a total loss of vegetation, as city dwellers would normally create and manage green spaces, some of which can be considered urban forests with a good level of tree cover [11].

Urban vegetation plays a multidimensional role by providing provisioning goods or services (food, fodder, medicine, and timber), regulating services (flood and erosion regulation, carbon storage, $CO_2$ sequestration, and climate change mitigation), supporting services (nutrient cycling and water production), and spiritual or cultural services (aesthetic benefits) [12,13]. While Wang and Gao [14] find that biodiversity conservation is beneficial to ecosystem functioning and stability, as well as climate change mitigation, Gomez-Baggethun and Barton [15] believe that urban parks and reserves are essential components of urban green infrastructure and play an important role in reducing the ecological footprint of urbanization. As a natural carbon sink, urban vegetation is essential to offset carbon dioxide ($CO_2$) emissions [14]. Urban trees influence air temperatures and building-energy consumption by fixing carbon during photosynthesis and storing carbon as biomass [16]. However, this depends on their composition, age, and biophysical growing conditions [6,17,18] as well as the density within the occupancy units [16,19,20]. For planning sustainable urban development, it is therefore necessary to recognize and integrate the structure, potential, and ability of urban trees to provide certain ecosystem functions and services to the urban environment and society. Carbon storage and sequestration are distinctive functions of plant diversity [6]. Plants accumulate woody biomass according to their tissue structure as they grow and generally sequester carbon dioxide ($CO_2$) in above- and belowground biomass at different rates [21]. Early attempts to estimate urban forest carbon stocks included studies by Dorney et al. [22], who used allometric models to quantify urban forest carbon stocks in a suburb of the city of Milwaukee. Since then, urban forest carbon stock studies have become increasingly common around the world [11].

In Benin, a few scientific studies have focused on the structure and carbon storage potential of urban species. The work of Sehoun et al. [23] was concerned with the study of species diversity in the green spaces of the city of Cotonou; those of Teka et al. [24] and Aziz [25] addressed the effects of trees and the alignment of seedlings on the local microclimate of the city. Orou et al. [7] assessed the diversity and structure of woody vegetation in the city of Malanville in North Benin. Atchadé et al. [26], on the other hand, was interested in studying plant diversity in different land use units in the city of Cotonou, and Amontcha [27] investigated public green spaces' species richness in Porto-Novo. Speaking of the evaluation of carbon stock by urban tree species, a few authors, such as Dangulla et al. [6], Vroh et al. [28], Du Toit et al., Folega et al. [19], Nero and Callo-concha [29], and Dangulla et al. [6], investigated urban tree biomass and carbon stock in Africa. At the time, in the United States and other developed countries, most of the studies were conducted on urban tree ability to store aboveground carbon [30–33]. According to Tang et al. [34], our understanding of the dynamics of urban carbon reservoirs, and specifically the dynamics of urban forest carbon stocks, remains limited. A good understanding of urban forest carbon stocks is essential for sustainable carbon management, which goes hand in hand with climate action and sustainable cities [11]. In this regard, understanding the structure and carbon storage ability of trees in African cities will therefore provide an opportunity to fill the knowledge gap on the distribution of carbon stocks within cities, as well as in the land use types from which cities were built. As a decision-making tool, this will facilitate the orientation of urban planning policies of cities to enrich green infrastructure with sustainable carbon footprints, which also catalyzes the path towards the green, inclusive, resilient, and sustainable African cities as envisioned by the United Nations Sustainable Development Goals (SDG11), as well as the aspirations of Agenda 63, The Africa We Want. In this line, the main objectives addressed by this research are to:

- Assess the structural parameters of trees in different urban land use units of Cotonou city;

- Determine the carbon stocks in the biomass of trees in the land use units of the city of Cotonou.

This is important in connection with the problem of climate change and differences in land use types, which can help the city management improve the situation. By doing so, this work can replace the role of urban vegetation in the adaptation and, especially, the mitigation of climate change in Cotonou, a metropolitan city in Benin, West Africa.

## 2. Methodology

### 2.1. Area of Study

The city of Cotonou is in the south of the Republic of Benin, between 6°20′ and 6°23′ north latitude and 2°22′ and 2°30′ east longitude. It is bordered to the north by Lake Nokoué, to the west by the municipality of Abomey-Calavi, to the east by the municipality of Sèmè-Kpodji, and to the south by the Atlantic Ocean (Figure 1). The city covers an area of 79 km$^2$ [35].

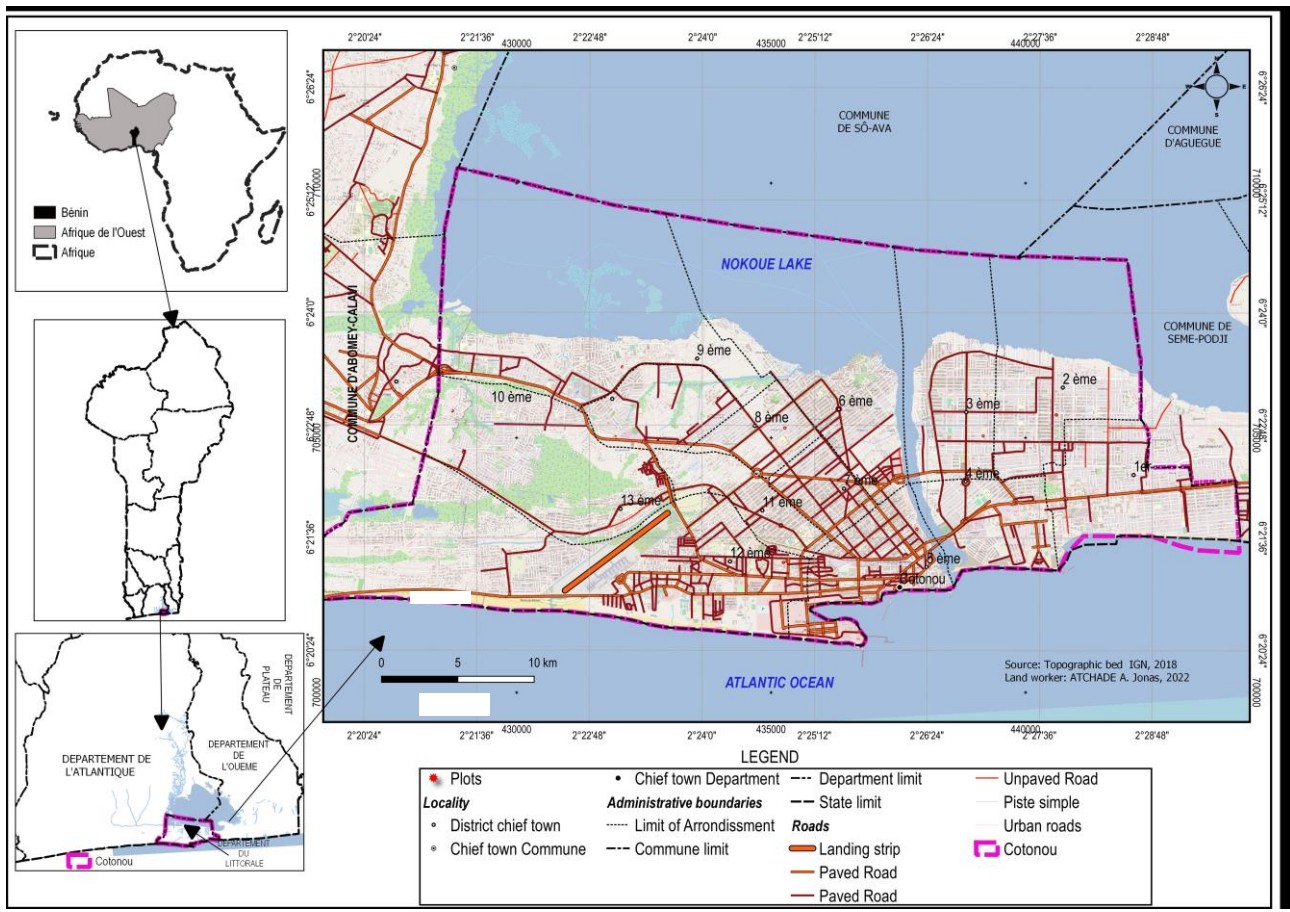

**Figure 1.** Location map of the research area.

Administratively, the city of Cotonou comprises 13 boroughs subdivided into 144 neighborhoods. Its population is 679,012 according to the general population and housing census [36]. The climate is humid subequatorial, with two dry seasons (mid-November to mid-March and mid-July to August) and two rainy seasons (mid-March to mid-July and September to mid-November). The average annual rainfall is 1200 mm, with 700–800 mm in the long rainy season and 400–500 mm in the short rainy season [37]. The average temperature in the coastal zone is 26.8 °C, with extremes of 36.6 °C and 16.5 °C. The average relative humidity in Cotonou is 84%. The hydrographic network consists of Lake Nokoué and the Atlantic Ocean. The types of soil encountered include sandy soils, ferruginous soils, and hydromorphic soils [23]. All these characteristics favor plant devel-

opment. The current urban matrix of the city offers a wide range of types of man-made and natural environments and vegetation ranging from totally unvegetated environments in the city centers to wooded private parks in residential areas to spontaneous vegetation in abandoned estates in the neighborhoods to fallows, plantations, ponds, marshes, and swamps in the peripheral areas of the city [38].

### 2.2. Data Collection

After having mapped the city of Cotonou (Figure 2), we randomly selected 8 of the 13 boroughs in the city to cover all the localities (central and peripheral districts).

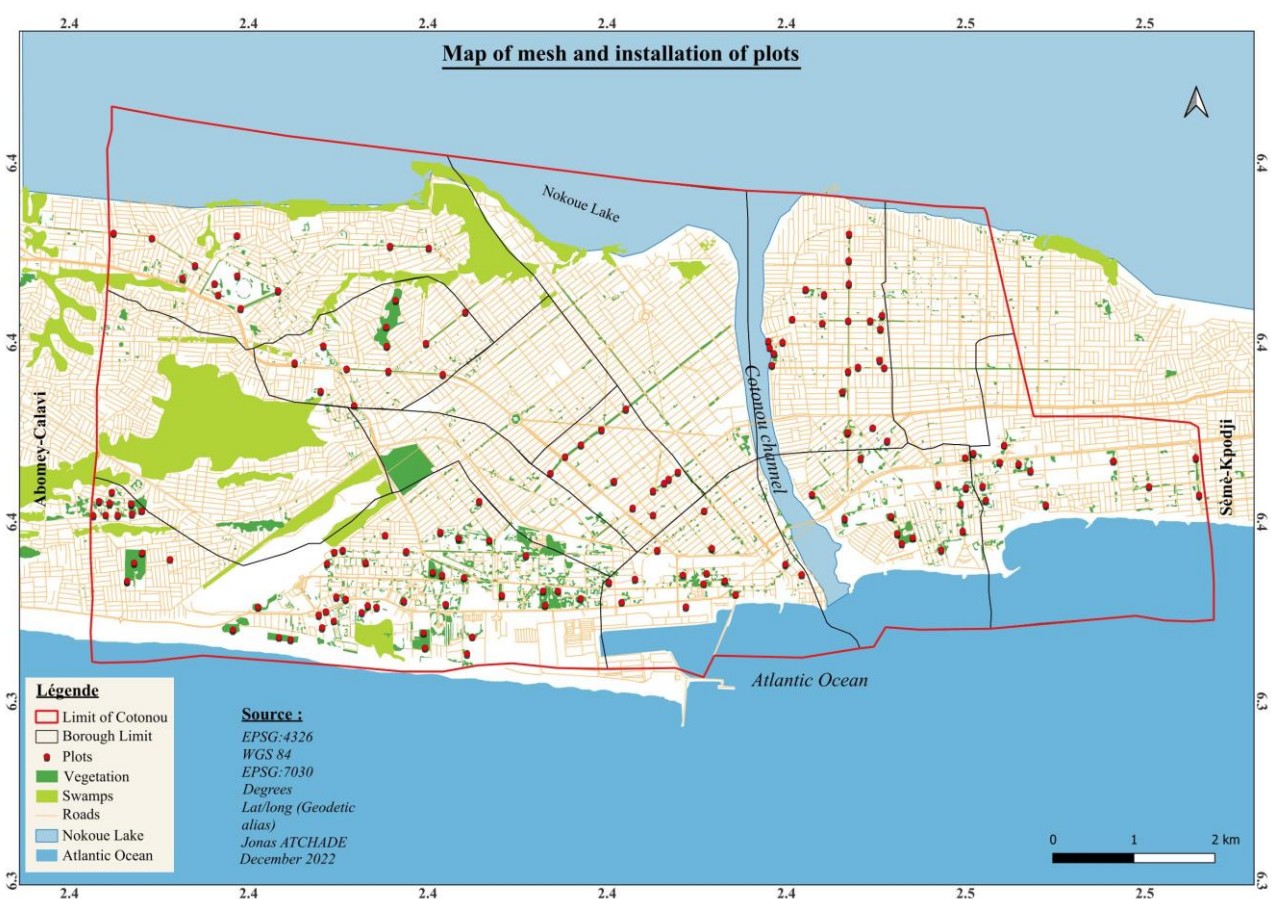

**Figure 2.** Inventoried tree map in the research area.

Based on the FAO definition [38] of urban forest, the study area was stratified into six land use types that correspond to urban forests in Cotonou: (1) commercial areas, including markets, stores, boutiques, restaurants, and vehicle repair shops; (2) roads covering main streets and boulevards and that are located in larger planting areas near roads; (3) residential areas covering houses, mosques, and churches; (4) schools covering private and public training and learning institutions, such as elementary schools, secondary schools, universities, schools, and vocational training centers; (5) administrative areas such as public and private utilities; (6) forest areas consisting of urban agricultural plots, urban green spaces, agroforestry systems, wetlands, irrigated agricultural land, and botanical gardens. Given the occupancy units, 149 rectangular and square plots of 25 m × 100 m and 50 m × 50 m were operated on following the guidelines of Thiombiano et al. [39] for forest inventory in Africa. In each of the plots, a systematic inventory of woody species was made on the one hand. On the other hand, the tree diameter at breast height (1.3 m) (Dbh ≥ 10 cm) and height (H ≥ 2 m), as well as the number of stems of each tree, were assessed.

*2.3. Statistical Analysis*

To study tree structure, 10 cm size classes based on Dbh (10–20 cm, 20–30 cm, 30–40 cm, 40–50 cm, 50–60 cm, 60–70 cm, 70–80 cm, 80–90 cm, 90–100 cm, 100–110 cm) and 2 m classes based on height (2–4 m, 4–8 m, 8–12 m, 12–14 m, 14–16 m, 16–18 m, 18–20 m, 20–24 m) were used, as applied by Folega et al. [19]. Therefore, the distribution of diameter and height classes was determined overall and by occupancy units. The observed distribution of diameter and height classes was fitted to the theoretical Weibull distribution [40]. The shape parameter was used to judge population status [41]. A log-linear analysis was performed to assess the overall fit between the observed and theoretical distributions (the Weibull distribution), as performed by Glele and Mette [42]. The fit of the parameters to the Weibull distribution allow for a better characterization of the variability of forest stand structure shapes.

To highlight the allometric relationship of total height–Dbh, aboveground biomass and diameter (AGB–Dbh), and aboveground biomass and height (AGB–height) of tree species across occupancy units, we used Standardized Major Axis regression (SMA) with the sma function of the smatr package [43]. These models allowed us to compare species' abilities to grow tall and store carbon across occupancy types.

Since biomass estimation using species-specific equations has not been possible due to the paucity of species-specific equations in the literature [44], especially in urban areas where woody species diversity is not negligible [45], we critically searched the literature to identify recent allometric models potentially suitable for aboveground biomass (AGB) estimation in West Africa. The generic allometric equation developed by Chave et al. [46] and recently used by Dangulla et al. [6] and Moussa et al. [11] was found to meet the objectives of aboveground biomass estimation. This equation is as follows:

$$AGB(kg) = 0.0673 \left( \rho D^2 H \right)^{0.976} \tag{1}$$

where AGB = aboveground biomass in kg,

$\rho$ = density of wood (g cm$^{-3}$),

D = diameter in cm at breast height (1.3 m), here Dbh, and

H = total tree height (m), which are used by the authors when destructive methods are not applicable, as in the case of urban forests [44]. Wood density was obtained from the global wood density database [47]. For species without wood density data, genus or family wood density was used. The model developed by Chave et al. [46] has a diameter range of 5−212 cm. The carbon stock in the different occupancy units was obtained by multiplying AGB by 0.47 [48]. The BIODIVERSITY package was used to obtain the outputs from the R.4.1.2 software.

To compare the structural parameters (total height, Dbh, basal area) and the carbon stock according to the type of land use, analyses of variance were performed. This was preceded by the Shapiro–Wilk and Levene tests, which, respectively, verified the residual normality and homogeneity. Generalized linear regression (negative binomial fit) was performed to assess the effect of land use units. For all models, Student–Newman–Keuls mean-structuring tests were performed under the library used by Mendiburu [49] to discriminate occupations. The level for significance was determined at $p < 0.05$. A one-way ANOVA test was used to test the differences between the mean values of the structural parameters, H, Dbh, and carbon stocks, for all the occupation units of the city.

The data collected were first processed and formatted using Microsoft 365's Excel spreadsheet. And then, all analyses were performed in R.4.1.2 software.

## 3. Results

*3.1. Diameter and Height Structures of Woody Plants*

Figure 3 presents the diameter (G) and height (H) structure of all species with adjustments of the Weibull distribution parameters in the city of Cotonou.

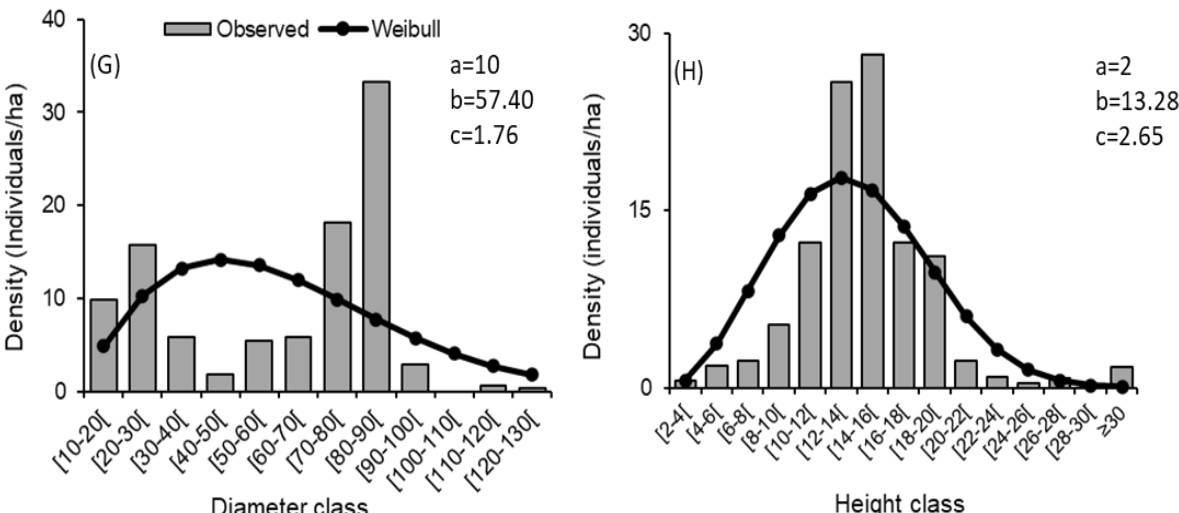

**Figure 3.** Diameter (G) and height (H) structures of all species with overlay of the Weibull distribution in Cotonou city. (a) is the position parameter (in cm), (b) is the average diameter, and (c) is the shape parameter related to the diameter structure in question. According to Husch et al. (2003), the Weibull can take several forms depending on the value of the shape parameter.

It can be noted from Figure 2 that the diameter structure of the trees inventoried in 149 plots in the city of Cotonou reveals the presence of individuals of all diameter classes, varying between 10 cm and 130 cm.

Individuals with diameters between 80 cm and 90 cm are the most numerous per hectare, followed by individuals of diameter classes 70–80 cm and 20–30 cm, respectively. It is rare to observe more than four trees per hectare for individuals with diameters over 90 cm. The least common individuals in the city are those of diameter class 110–120 cm and 120–130 cm and average only one per hectare.

The height distribution of the trees in a stand has an overall Gaussian shape that is right skewed. The shape parameter c is equal to 2.65, ranging from 1 to 3.6. Urban plant species with height between 14 m and 16 m are the most numerous per hectare. The modal class is the same class of dominant heights. It is very rare to find five individuals above 20 m in height per hectare in the whole city of Cotonou.

### 3.2. Similarity of Allometric Relationships and Aboveground Biomass Mobilization across Type of Land Use

Table 1 summarizes the results of Standardized Major Axis regression (SMA) comparing the allometric relationship between total height and diameter (height vs. Dbh), the aerial biomass and diameter relationship (AGB vs. Dbh), and the aerial biomass and height relationship (AGB vs. height) of woody species according to type of land use.

Referring to Table 1, we see an overlap between the lower (Liminf) and upper (Limsup) limits, that is, between the confidence intervals across some of type of land use. Taking this into account, it implies that the allometric relationship between height and diameter (H vs. Dbh) of urban trees of the same species is statistically similar in type of land use, such as administrative (Slope = 0.48; Liminf = 0.38; Limsup = 0.61; $p < 0.001$), commercial (Slope = 0.27; Liminf = 0.18; Limsup = 0.39; $p < 0.001$), establishments (Slope = 0.46; Liminf = 0.41; Limsup = 0.53; $p < 0.001$), residences (Slope = 0.45; Liminf = 0.38; Limsup = 0.52; $p < 0.001$), and roads (Slope = 0.51; Liminf = 0.44; Limsup = 0.59; $p < 0.001$). In other words, the way in which tree heights change with diameter in administrative areas is the same in commercial areas, settlements, residential areas, and roads. In contrast, this relationship is statistically different in the green spaces (Slope = −0.35; Liminf = −0.47; Limsup = −0.25; $p < 0.001$) of the city. Note also that the ability of these trees to grow in height in these five types of land use of the city is positively strongly influenced by their diameter ($R^2 = 0.99$).

**Table 1.** Summary of Standardized Major Axis regression (SMA) results comparing the total allometric height vs. Dbh, AGB vs. Dbh, and AGB vs. height relationships of tree species according to type of land use.

| Type of Land Use | Statistics | Height vs. Dbh | | AGB vs. Dbh | | AGB vs. Height | |
|---|---|---|---|---|---|---|---|
| | | Elevation | Slope | Elevation | Slope | Elevation | Slope |
| Administrative | Estimate | 0.36 | 0.48 | −3.64 | 2.00 | −5.15 | 4.16 |
| | Liminf | 0.17 | 0.38 | −3.71 | 1.95 | −6.30 | 3.30 |
| | Limsup | 0.56 | 0.61 | −3.56 | 2.04 | −4.00 | 5.26 |
| | | $R^2 = 0.99$ | $p < 0.001$ | $R^2 = 0.99$ | $p < 0.001$ | $R^2 = 0.01$ | $p = 0.339$ |
| Commercial area | Estimate | 0.72 | 0.27 | −3.65 | 1.96 | −9.00 | 7.39 |
| | Liminf | 0.57 | 0.18 | −3.71 | 1.92 | −12.20 | 5.09 |
| | Limsup | 0.88 | 0.39 | −3.58 | 2.00 | −5.81 | 10.73 |
| | | $R^2 = 0.99$ | $p < 0.001$ | $R^2 = 0.99$ | $p < 0.001$ | $R^2 = 0.03$ | $p = 0.362$ |
| Green spaces | Estimate | 1.77 | −0.35 | −3.61 | 1.96 | 6.48 | −5.69 |
| | Liminf | 1.57 | −0.47 | −3,69 | 1.92 | 4.33 | −7.81 |
| | Limsup | 1.97 | −0.25 | −3.52 | 2.01 | 8.62 | −4.14 |
| | | $R^2 = 0.99$ | $p < 0.001$ | $R^2 = 0.99$ | $p < 0.001$ | $R^2 = 0.00$ | $p = 0.583$ |
| Establishments | Estimate | 0.30 | 0.46 | −3.83 | 2.08 | −5.19 | 4.51 |
| | Liminf | 0.19 | 0.41 | −3.90 | 2.05 | −5.87 | 3.94 |
| | Limsup | 0.41 | 0.53 | −3.77 | 2.12 | −4.50 | 5.15 |
| | | $R^2 = 0.99$ | $p < 0.001$ | $R^2 = 0.99$ | $p < 0.001$ | $R^2 = 0.25$ | $p = 0.060$ |
| Residential area | Estimate | 0.45 | 0.44 | −3.74 | 2.04 | −5.83 | 4.62 |
| | Liminf | 0.34 | 0.38 | −3.80 | 2.01 | −6.69 | 3.93 |
| | Limsup | 0.57 | 0.52 | −3.68 | 2.08 | −4.98 | 5.44 |
| | | $R^2 = 0.99$ | $p < 0.001$ | $R^2 = 0.99$ | $p < 0.001$ | $R^2 = 0.02$ | $p = 0.095$ |
| Roads | Estimate | 0.19 | 0.51 | −3.89 | 2.14 | −3.89 | 4.17 |
| | Liminf | 0.06 | 0.44 | −3.94 | 2.11 | −3.94 | 3.62 |
| | Limsup | 0.32 | 0.59 | −3.83 | 2.17 | −3.83 | 4.80 |
| | | $R^2 = 0.99$ | $p < 0.001$ | $R^2 = 0.99$ | $p < 0.001$ | $R^2 = 0.46$ | $p = 0.000$ |

Liminf and Limsup are, respectively, the lower and upper limits of the 95% confidence intervals, estimate is the regression coefficient, and $R^2$ is the coefficient of determination. For the same species, the overlap of the confidence intervals (Liminf and Limsup) around the slopes indicates that these relationships are statistically similar. Slope is the slope; elevation is the intercept. H is total height, Dbh is diameter at breast height, and AGB is aboveground biomass.

Similarly, Table 1 outlines the AGB vs. Dbh relationship and indicates that the ability of woody plants to mobilize aboveground biomass and thus store carbon is statistically significantly influenced by their diameter ($R^2 = 0.99$; $p < 0.001$) in all land use units. This ability to store biomass as a function of the diameter of trees of the same species is significantly similar in type of land use, such as administrative (Slope = 2; Liminf = 1.95; Limsup = 2.04; $p < 0.001$), commercial (Slope = 1.96; Liminf = 1.92; Limsup = 2; $p < 0.001$), and green space (Slope = 1.96; Liminf = 1.92; Limsup = 2.01; $p < 0.001$) areas of the city. Apart from this first group, this similarity in the ecological function of trees of the same species is statistically significantly the same in other types of land use, such as settlements (Slope = 2.08; Liminf = 2.05; Limsup = 2.12; $p < 0.001$), residences (Slope = 2.04; Liminf = 2.01; Limsup = 2.08; $p < 0.001$), and roads (Slope = 2.14; Liminf = 2.11; Limsup = 2.17; $p < 0.001$). Further, these results suggest that for trees of the same diameter, those planted along roads have a greater capacity to mobilize biomass and therefore store carbon than those found in the other land use units (slope = 2.14).

Regarding the ability to mobilize biomass and carbon storage as a function of tree height (AGB vs. height), Table 1 shows that there is an overlap between the confidence intervals of some of the types of land use, and this implies that this ecological ability

of trees of one species is similar but statistically insignificant in the occupancy units used as administrative areas (Slope = 4.16; Liminf = 3.30; Limsup = 5.26; *p* = 0.339), commercial areas (Slope = 7.39; Liminf = 5.09; Limsup = 10.73; *p* = 0. 362), establishments (Slope = 4.51; Liminf = 3.94; Limsup = 5.15; *p* = 0.000), residences (Slope = 4.62; Liminf = 3.93; Limsup = 5.44; *p* = 0.095), and roads (Slope = 4.17; Liminf = 3.62; Limsup = 4.80; *p* = 0.000). However, this behavior relative to the ecological function of urban tree species is different in green spaces (Slope = −5.69; Liminf = −7.81; Limsup = −4.14; *p* = 0.583). It should also be noted that the share of tree height in the variability of its capacity to mobilize biomass and thus store carbon is very insignificant in administrative areas ($R^2$ = 0.01), commercial areas ($R^2$ = 0.03), residential areas ($R^2$ = 0.02), and green areas ($R^2$ = 0.00).

### 3.3. Graphical Similarity of Allometric Relationships According to Type of Land Use

Referring to Figure 4, the results indicated overall statistically similar lines for the different types of land use with respect to the relationship between tree height and Dbh (livelihood ratio, LR = 13. 88; degree of freedom, Df = 5; *p* = 0.016) and between diameter and aboveground biomass and, thus, carbon stock (LR = 66.68, Df = 5, *p* ≤ 0.001), and not significant between height and aboveground biomass and, thus, carbon stock (LR = 10.42, Df = 5, *p* = 0.064).

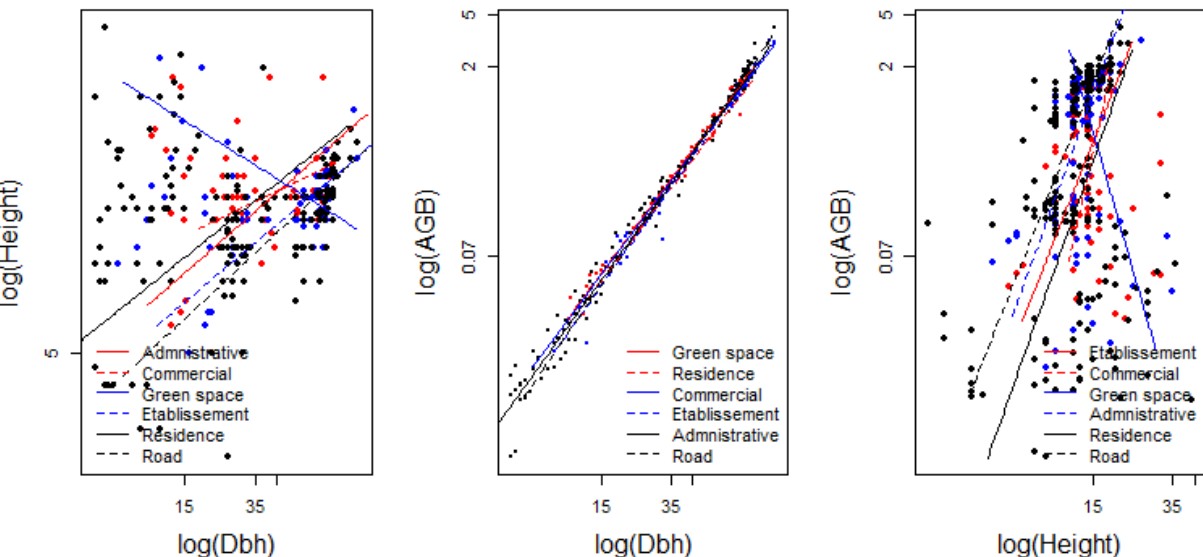

**Figure 4.** The evolution of the relationship between total height and Dbh, AGB and Dbh, and AGB and height of the species according to the land use units in the city of Cotonou.

From this figure, the parallelism between the lines justifies the equality between the slopes. Thus, the total allometric relationship of height and Dbh of plants of the same species is the same across the different occupation units apart from the green spaces of the city of Cotonou. This observation is valid in all the occupation units except for the green spaces when it comes to the relationship between the ability to mobilize aerial biomass and the total height of trees (AGB vs. height). In contrast, the linear regression lines between aboveground biomass mobilization and tree diameter (AGB vs. Dbh) showed not only the very close relationship between the two variables but also the similarity of this ecological function in all land use units without exception. This expression of a close link between tree diameter and aboveground biomass mobilization reinforces the choice of the allometric equation used for the estimation of biomass and carbon stock in the occupancy utilities of the city of Cotonou.

In other words, the way in which tree height evolves with diameter in administrative areas is virtually the same in commercial areas, settlements, residential areas, and roads. On the other hand, this relationship is different in a city's green spaces. Furthermore, we

also note that the ability of these tree species to grow in height in administrative areas is virtually the same in commercial areas, settlements, residential areas, and city roads, and is strongly influenced by their diameter.

### 3.4. Structural Parameters and Aerial Carbon Density in the Different Land Use Units

Table 2 presents the mean values and Standard Deviations (SD) of the structural parameters (basal area (G), density (N) of trees per hectare, diameter at breast height (Dbh), height (H) of trees) and carbon stock (AGC) across land use units. Overall, the results showed that the mean values of the different structural parameters and the carbon stock rate varied significantly across the land use units of the city ($p < 0.001$).

**Table 2.** Inferential tests and descriptive statistics of structural parameters and carbon stock in the land use units of the city of Cotonou.

| Type of Land Use | G ($m^2$ $ha^{-1}$) | | N (feet $ha^{-1}$) | | Dbh (cm) | | H (m) | | AGC $\times 10^3$ (kg $ha^{-1}$) | |
|---|---|---|---|---|---|---|---|---|---|---|
| | Mean | ET | Mean | ET | Mean | ET | Mean | ET | Mean | ET |
| Administrative area | 3.00 [b] | 4.91 | 10.0 [d] | 8.73 | 54.91 [b] | 28.14 | 15.20 [a] | 5.09 | 0.88 [bc] | 0.72 |
| Commercial area | 2.39 [b] | 3.31 | 15.00 [b] | 13.48 | 39.25 [c] | 22.54 | 13.6 [ab] | 2.03 | 0.39 [d] | 0.47 |
| Green spaces | 7.37 [a] | 8.35 | 15.60 [b] | 14.54 | 70.78 [a] | 27.05 | 14.96 [ab] | 4.89 | 1.21 [a] | 0.63 |
| Establishments | 5.65 [ab] | 4.75 | 13.87 [c] | 7.68 | 67.30 [a] | 25.06 | 13.68 [ab] | 3.16 | 1.10 [ab] | 0.62 |
| Residential area | 2.69 [b] | 3.74 | 9.60 [d] | 6.76 | 44.17 [c] | 31.94 | 13.84 [ab] | 5.83 | 0.65 [c] | 0.73 |
| Roads | 7.56 [a] | 6.23 | 18.73 [a] | 11.34 | 65.19 [a] | 26.61 | 12.91 [b] | 3.31 | 1.19 [a] | 0.76 |
| Globale | 4.5 | 5.24 | 12.72 | 9.42 | 57.94 | 29.71 | 13.86 | 4.42 | 0.94 | 0.73 |
| Fvalue/Deviance | 5.02 | | 170.47 | | 17.33 | | 2.91 | | 14.96 | |
| Probability | <0.001 | | <0.001 | | <0.001 | | 0.01 | | <0.001 | |

In the column, means followed by the same letters are statistically similar at the 5% threshold and those with different letters are statistically different. F value and deviance represent, respectively, the Fisher F statistics of the ANOVA models and the residual deviance from the generalized linear regression with binomial negative fit. Mean: average, SD: standard error, N: density, G: basal area, Dbh: diameter at breast height, H: height, and AGC: Above-Ground carbon stock.

In general, the basal area of trees in the city of Cotonou is estimated at $4.52 \pm 5.24$ $m^2$ $ha^{-1}$, with an average of 13 trees per hectare ($12.72 \pm 9.42$ feet $ha^{-1}$). Among the occupation units, green spaces ($7.37 \pm 8.35$ $m^2$ $ha^{-1}$) and roads ($7.56 \pm 6.32$ $m^2$ $ha^{-1}$) contain trees that contributed more to the city's land surface with, respectively, 16 ($15.60 \pm 14.54$ feet $ha^{-1}$) and 19 ($18.73 \pm 11.34$ feet $ha^{-1}$) trees per hectare on average. Residential ($2.69 \pm 3.74$ $m^2$ $ha^{-1}$), commercial ($2.39 \pm 3.31$ $m^2$ $ha^{-1}$), and administrative ($3.00 \pm 4.91$ $m^2$ $ha^{-1}$) areas have the lowest tree basal areas, with an average of 15 feet/ha in commercial areas and about 10 feet of trees per hectare in administrative and residential areas (Table 2). Basal area in the latter three occupancy units is statistically similar ($p > 0.05$) but varies significantly from that in green spaces, roads, and establishments ($p < 0.001$). No significant difference is observed between the land areas of trees sheltered by roads and green spaces. According to the same table, in terms of tree density, roads ($18.73 \pm 11.34$ feet $ha^{-1}$), green spaces ($15.60 \pm 14.54$ feet/ha), and commercial areas ($15.00 \pm 13.48$ feet/ha) recorded the highest number of trees per hectare, followed by settlements ($13.87 \pm 7.68$ feet $ha^{-1}$). The lowest densities were observed in administrations ($10.07 \pm 8.73$ feet $ha^{-1}$) and residences ($9.60 \pm 6.76$ feet $ha^{-1}$), with no statistically significant difference ($p > 0.05$). There is also no significant difference between the tree density of green spaces and commercial areas, which is not the case with the roadways. The average diameter of trees in the entire city of Cotonou is estimated at $57.94 \pm 29.71$ cm. Among the occupancy units, roads ($65.19 \pm 26.61$ cm), green spaces ($70.78 \pm 27.05$ cm), and establishments ($67.30 \pm 25.06$ cm) have the largest tree diameters in the city of Cotonou. There is no significant difference in the mean diameters of trees in these occupancy units (Table 2). However, small-diameter trees are found in the residences ($44.17 \pm 31.94$ cm) and commercial areas ($39.25 \pm 22.54$ cm) of the city. From Table 2, in Cotonou, trees reached an average height of $13.86 \pm 4.42$ m. The tallest trees are found

in the administrations (15.20 ± 5.09 m), while those with short heights are sheltered by the roadway (12.91 ± 3.31 m). There is no statistically significant difference ($p > 0.05$) between the average height of trees sheltered by commercial areas (13.65 ± 2.03 m), green spaces (14.96 ± 4.89 m), establishments (13.68 ± 3.16 m), and residences (13.84 ± 5.83 m). From the same table, it appears that the amount of carbon sequestered by urban vegetation in the city of Cotonou varies statistically significantly from one occupancy unit to another, except between roads and green spaces. About 1000 kg of carbon ($0.94 ± 0.73 × 10^3$ kg ha$^{-1}$) is stored per hectare by urban vegetation in the city of Cotonou. The highest carbon stocks are sequestered by green space vegetation (1.21 ± 0.63 t ha$^{-1}$), followed by road vegetation ($1.19 ± 0.76 × 10^3$ kg ha$^{-1}$), with no statistically significant difference between the two land use units ($p > 0.05$). The lowest carbon stock was recorded in trees in shopping malls ($0.39 ± 0.47 × 10^3$ kg ha$^{-1}$). There was a statistically significant difference in the amount of carbon stored by plants in administrative areas ($0.88 ± 0.72 × 10^3$ kg ha$^{-1}$), establishments ($1.10 ± 0.47 × 10^3$ kg ha$^{-1}$), residential areas ($0.65 ± 0.47 × 10^3$ kg ha$^{-1}$), and commercial areas ($0.39 ± 0.47 × 10^3$ kg ha$^{-1}$) (Table 2).

*3.5. Species with High Carbon Storage Potential*

Table 3 shows the top 10 species with the highest carbon storage potential in the city of Cotonou. From the analysis of the table, it appears that *Khaya senegalensis* ($1.49 ± 0.06 × 10^3$ kg ha$^{-1}$), a native species of the Meliaceae family, followed by *Mangifera indica* ($1.49 ± 0.06 × 10^3$ kg/ha), an exotic species of the Anacardiaceae family, are the species with the highest potential for aerial carbon storage in Cotonou. The tenth and last species of this class of tree is *Acacia auriculiformis*, an exotic species of the Fabaceae family with a carbon storage capacity of $0.26 ± 0.03 × 10^3$ kg ha$^{-1}$. From the same table, it is noted that this class of species leaders in carbon storage in the city of Cotonou is 90% exotic species, and that the family Meliaceae, Combretaceae, and Fabaceae are, respectively, the leading families.

**Table 3.** Species with high potential for carbon storage in the city of Cotonou.

| Species | Family | Mean AGC × $10^3$ (kg ha$^{-1}$) | SE |
|---|---|---|---|
| *Khaya senegalensis* [1] | Meliaceae | 1.49 | 0.06 |
| *Mangifera indica* [2] | Anacardiaceae | 1.44 | 0.04 |
| *Terminalia mantaly* [2] | Combretaceae | 1.35 | 0.06 |
| *Leucaena leucocephala* [2] | Fabaceae | 1.23 | 0.23 |
| *Terminalia cattapa* [2] | Combretaceae | 1.18 | 0.05 |
| *Azadirachta indica* [2] | Meliaceae | 0.56 | 0.20 |
| *Tectona grandis* [2] | Verbenaceae | 0.55 | 0.00 |
| *Casuarina equisetifolia* [2] | Casuarinaceae | 0.51 | 0.36 |
| *Polyalthia longifolia* [2] | Annonacées | 0.30 | 0.05 |
| *Acacia auriculiformis* [2] | Fabaceae | 0.26 | 0.03 |

SE: Standard error, aboveground biomass carbon stock (AGC). [1] native species, [2] exotic species.

## 4. Discussion

*4.1. Dendrometry Structure of Trees*

In this diametric structure of the trees in the city of Cotonou, a predominance of individuals in the 80–90 cm diameter class per hectare was recorded. The results of the work of Lahoti et al. [17] showed that the 60–90 cm diameter class abounds in many species in the urban tropical vegetation of Nagpur (India). The same author inventoried more than 200 individuals of a diameter class of above 90–120 cm in an urban environment. In contrast, work by Moussa et al. [11] in two cities in the Sahel (Niamey and Maradi) resulted in 22.75–40 cm being the dominant diameter class. This could be explained by the fact that the annual rainfall and the state of soil degradation in the Sahel are not as favorable for the growth of some urban plant species as in cities located on the coast such as Cotonou. The diameter structure of plant species in the city of Cotonou has a "bell" shape. For Husch et al. [41], this structure with the bell shape parameter should fit a normal distribution, for

an asymmetry (right-hand side here) can be related to the living conditions of the trees. The parameter c of the Weibull distribution taking a value of 1.76 in the city of Cotonou is, in this case, between 1 and 3.6 and indicates a right-handed asymmetry, posing, all other things being equal, a problem of recruitment of young individuals into the classes of old individuals. This constitutes a decision support tool in the ecological management of the territory. The height structure of the trees in the city of Cotonou is generally like an asymmetric distribution. Urban plant species with heights between 14 m and 16 m are the most numerous per hectare. Folega et al. [19] revealed height classes ranging from 2 m to 24 m in the city of Atakpamey. The work of Dangulla et al. [6] showed that trees in a city in northwestern Nigeria (Zaria), a country bordering Benin, have varying heights of between 8 m and 20 m. Although the diameter and height classes indicate the structure of a vegetation, it should also be noted that the shape of the diameter structure of individuals varies according to the type of occupation, the living conditions of the trees, the species present, and their age. It should also be worth adding that many differences in urban tree vegetation are mainly due to anthropogenic activities.

*4.2. Allometric Relationships and Aboveground Biomass Mobilization across Urban Types of Land Use*

The results indicated overall statistically similar lines for the different types of occupations, with respect to the relationship of significance between tree height and Dbh and between diameter and aboveground biomass (thus carbon stock), and nonsignificance between height and aboveground biomass. In their investigation of the influence of biophysical parameters in the mobilization and storage of aerial carbon by urban trees, Wang and Gao [14] came to the conclusion that between the diameter and height of trees, the diameter influences more the mobilization of aerial biomass at the species level depending on the land use units on which the trees are installed. By the same logic, allometric relationships between dendrometric parameters, in this case diameter, largely determine the amount of carbon a plant can mobilize in biomass, even if there are other underlying factors [5]. Our results, while pointing to an $R^2 = 0.99$ (AGB vs. Dbh) value across all land use units (Table 1), agree with those of these authors.

*4.3. Structural Parameters and Tree Aboveground Carbon Storage Potential*

In the city of Cotonou, overall, the results showed that the mean values of the different structural parameters and the rate of aboveground carbon stock varied significantly across the city's land use units ($p < 0.001$). Vegetation structure is described by tree density, basal area, and height and diameter class distribution. The average tree density and basal area of the Cotonou urban forest deviate from what is reported by Nero and Callo-Concha [20] for urban forests and plantations in the city of Kumasi, Ghana. The average basal area and density of trees in the Kumasi urban forest are 55.5 $m^2$ $ha^{-1}$ and 377 trees $ha^{-1}$, respectively. This discrepancy can be attributed to the conservation or protection of large trees in natural forest relics, cemeteries, public parks, institutional complexes, and agricultural lands in Kumasi. However, the density and basal area of trees in Cotonou city roads (18.73 trees $ha^{-1}$ and 7.6 $m^2$ $ha^{-1}$) are higher than those in Malanville city roads (12.09 trees $ha^{-1}$ and 1.75 $m^2$ $ha^{-1}$), as revealed by the work of Orou et al. [50,51]. The low density of trees in the land units of the city of Cotonou could be explained by several factors referencing Aziz et al. [52]. For these authors, the populations pointed fingers to anthropic actions ($p = 0.001$), management conditions ($p = 0.01$), organizational framework, and biophysical conditions as currently significant constraining factors for tree planting and survival in Benin. This low tree density could also be explained by the fact that some land use units in the city of Cotonou are full of many young species that our work is not interested in, as trees with Dbh greater than or equal to 10 cm were the inventory targets of our work. This argument is reinforced by the results of the new work of Aziz [25] which revealed the existence of 1034 young individuals divided into seven species and seven families at the level of the road in the city of Cotonou. Better still, the survival and develop-

ment of urban plant species are a function of biophysical parameters, different levels of maintenance, exposure to pollution, disturbance, and stress [17]. In addition, construction expansion in institutional compounds leads to the felling of old trees, as regularly reported in local media. The negligence of the authorities, the lack of rigorous policies, and the lack of monitoring tools mean that avenue plantings and green spaces in institutions are left to deteriorate. The lack of control tools leaves the avenue plantations and the institutional green spaces in a state of great vulnerability. The different units of occupancy therefore require greater attention in terms of preservation actions and compensatory planting efforts, as well as their rigorous monitoring.

Nero and Callo-Concha [20], Folega et al. [19], Moussa et al. [11], and Dangulla et al. [6] inventoried, in their works, more trees in urban green spaces, administrations, and roadways than in concessions. Our results are consistent with those of these authors in that the density of trees per hectare inventoried in urban green spaces, establishments, and roads are significantly higher than in residential areas of the city of Cotonou (Table 2).

Carbon sequestration rates varied significantly between tenure units ($p < 0.001$) (Table 2). This difference could be explained by the variation in tree diameters and densities, because for Folega et al. [19] and Moussa et al. [11], structural parameters, especially tree diameter and density, largely influence the carbon storage potential of urban species. The average value for the entire Cotonou commune is estimated at 0.94 t ha$^{-1}$ for about 13 trees per hectare, which is far from that (1.4 t ha$^{-1}$) reported by Vroh et al. [28] in the Plateau commune in Abidjan for a total of 233 trees. The amount of carbon stored by trees in settlements (1.1 t ha$^{-1}$) and roads (1.19 t ha$^{-1}$) is significantly higher than that recorded in residential (0.65 t ha$^{-1}$) and administration (0.88 t ha$^{-1}$) areas of the city of Cotonou. These results are consistent with those of Moussa et al. [11], who found that the carbon stored by establishments (31.77 t ha$^{-1}$) and roads (43.24 t ha$^{-1}$) is greater than that stored in residential (18.36 t ha$^{-1}$) and administrative (27.84 t ha$^{-1}$) areas of the city of Niamey. Similarly, the amount of carbon stored by the green spaces (1.21 t ha$^{-1}$) of Cotonou is greater than that of the green parks (0.7 t) of the Plateau commune of Abidjan. The difference between these carbon rates between cities is due to the structural parameters of the trees in the cities as well as between occupancy units within the same city. More importantly, carbon density is negatively associated not only with urban development intensity and social gradient, but also by the morphology of trees in different occupancy units of the city [8]. From a management perspective, it is important to emphasize that a lot of the city's tree biomass is stored in larger, older trees. This means that those trees need to be well cared for and that a younger generation of trees needs to be planted and maintained in anticipation of the older trees dying in the coming decades. This is very important for the ecological policy of cities across the country and the continent.

In this study, *Khaya senegalensis* and *Mangifera indica*, one indigenous and the other introduced from Asia as part of the colonial forestry program, have higher stem counts, biomass, and carbon stock. Other species, such as *Terminalia mentally*, *Terminalia cattapa*, *Azadirachta indica*, *Tectona grandis*, and *Polyalthia longifolia*, etc., did not fail to make the top 10 list of species with high carbon storage potential (Table 3). Some plant species are highly capable of storing carbon in urban environments, depending on biophysical conditions, age, resilience, and, especially, structural parameters [6]. There is also variability in the contribution of biomass and carbon stock by different species, as species differ in their total net productivity, proportion of above- and belowground carbon, and tissue turnover rates. Moussa et al. [11] and Dangulla et al. [6] found *Azadirachta indica*, *Mangifera indica*, *Khaya senegalensis*, *Parkia biglobosa*, and *Adansonia digitata* as the species with high carbon storage potential in the cities Niamey (Niger) and Zaria (Nigeria), respectively. These species, found in the city of Cotonou, can therefore be recommended in enhanced combination with other native species, such as *Khaya senegalensis*, *Parkia biglobosa*, and *Adansonia digitata*. This highlights the importance of species selection in urban tree planting and climate change mitigation programs for sustainability and the improvement of the ecological footprint of cities in Africa.

## 5. Conclusions

This work outlines tree structures and their ability to store aboveground carbon in the city of Cotonou. The results show variations in carbon stocks between land use units in the city. Currently, roads, green spaces, and establishments are the land use units that have recorded more aboveground carbon stock in the city of Cotonou. These variations in carbon stocks between occupancy types are explained by densities, tree age, or maturity (which were revealed by dendrometric parameters like diameter and heights), and other factors related to biophysical and anthropogenic constraints, management conditions, and organizational framework that this study does not highlight. *Khaya senegalensis*, *Mangifera indica*, and *Terminalia mentally* top the list of the top 10 species with high carbon storage potential. This raises the need to combine native and exotic species and to balance the presence of trees and forests for specific purposes in the city. This is the reason for insisting on reserving sufficient space for urban ecological infrastructure and human activities in all the different types of the land use of the city. Carbon stocks and species with high storage potential should be considered in urban planning and included in national and regional carbon budgets. These results add to the global carbon budget datasets and are relevant to urban climate change mitigation policy.

**Author Contributions:** Conceptualization, A.J.A.; Methodology, A.J.A. and A.A.D.; Software, A.J.A. and A.A.D.; Validation, K.A.; Formal analysis, A.J.A., S.A. and M.K.; Investigation, A.J.A.; Resources, M.K. and K.A.; Data curation, A.J.A. and S.A.; Writing—original draft, A.J.A.; Writing—review and editing, M.K. and F.F.; Visualization, M.K., F.F. and K.W.; Supervision, M.D., K.W. and K.A. All authors have read and agreed to the published version of the manuscript.

**Funding:** This research was funded by the Regional Centre of Excellence on Sustainable Cities in Africa (CERViDA_DOUNEDON), the Association of African Universities (AUA), and the World Bank Group. This paper is a part of PhD data collection results whose the amount is 4943.49 USD.

**Institutional Review Board Statement:** Not applicable.

**Informed Consent Statement:** Not applicable.

**Data Availability Statement:** Data will be made available on request.

**Conflicts of Interest:** The authors declare no conflict of interest.

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
