# Peer review of "Urban Flora Structure and Carbon Storage Potential of Woody Trees in Different Land Use Units of Cotonou (West Africa)"

_urbansci, doi:10.3390/urbansci7040106_

Round 1
Reviewer 1 Report
This fascinating article aims to study the structural parameters and carbon storage potential of trees in the land use units of the city of Cotonou in southern Benin. The findings of this work will contribute to better management of urban forest C at the local and regional levels. Overall, the paper is well-written and easy to follow (despite some minor typos).
Introduction: This is presented very well and summarizes the issue at hand and the current state of urban tree carbon storage.
Method:
- The quality of Figure 1 is low. Some of the text is difficult to read.
- Did you run a best-fit analysis? How do you justify the choice of the Weibull distribution? The Weibull distribution is not the best fit for the diameter class distribution (Figure 3a). Would a bimodal P. distribution work best in this situation?
Results and discussion: Well presented.
Reference: many formatting issues noted and missing references (e.g., #9)
Minor edits. There are a few typos throughout the manuscripts.
Author Response
Dear Reviewer
Thank you for taking the time to provide quality comments on our manuscript despite your busy schedule.
Taking into account your comments, please find the answers point by point in the
lines below.
Point by pont responses
Introduction
The reviwer approved that he has no concerns about the introduction.
Method:
Point1: The quality of Figure 1 is low. Some of the text is difficult to read.
Response 1: This was due to the size. We've enlarged the figure for greater visibility. You can see this in the revised manuscript.
Point 2: Did you run a best-fit analysis? How do you justify the choice of the Weibull distribution? The Weibull distribution is not the best fit for the diameter class distribution (Figure 3a). Would a bimodal P. distribution work best in this situation?
Response 2: According to Kakaï and Mette (2016), the establishment and interpretation of diameter structures, whatever the diameter classes of tree species, are essential for decision-making in urban or rural forest management. We didn't carry out a best-fit analysis, as the aim here is to highlight the appearance of tree diameter classes and see how they could fit with Weibull theory. Interpretation of this diameter class structure will give us an idea of the current ecological management options, and may help guide future tree planting decisions in the city. Already with this result, we can understand that the trees are of a diameter class both young and old, even old trees dominate the Urban stand. With this observation, we can already suggest to decision-makers the pressing need to continue greening the city with the installation of young plans.
Results and discussion
According to reviewer comments, results and discussion are Well presented.
Referneces
Point 3 : many formatting issues noted and missing references (e.g., #9)
Response 3 : This isssu is addressed. You can look at it in the revised version of the manuscript submitted.
Thanks for taking time to make your observations which allow us to improve the manuscript.

Reviewer 2 Report
This paper presented an interesting design of research and findings. Carbon and trees in cities is quite important for human being's social development.
while, there are some minor issues should be noted
In the paper, the data such as 0.94.103 kg ha-1 shoud be 0.94×103 kg ha-1
the references style is non-standard. and some different syles occurred.
In figure 3, the mean of a, b,c should mentioned in caption.
In section 3.2, the words of results should be slimed down because the main results in presented in Table 1.
In Table 3, native or exotic species could be marked.
the English expression should be improved, and I found that there are some minor mistakes should be corrected. such as:
Line 80, few should be "a few"
line 87, only sould be deleted.
Line 93, "For Tang, Chen, & Zhao, (2016)", this kind of use should be in the style of citation. the authours used this kind of expression for figures, tables and citations some times.
Author Response
Dear Reviewer
Thank you for taking the time to provide quality comments on our manuscript despite your busy schedule.
Point-by-point response
Point 1: In the paper, the data such as 0.94.103 kg ha-1 shoud be 0.94×103 kg ha-1
Response 1: This is addressed.
Point 2: the references style is non-standard. and some different syles occurred.
Response 2: This is addressed
Point 3: In figure 3, the mean of a, b,c should mentioned in caption.
Response 3: This is addressed in the revised manuscript.
a means position parameter (in cm), b is the average diameter and c is the shape parameter related to the diameter structure in question. According to Husch et al (2003), the Weibull can take several forms depending on the value of the shape parameter.
Point 4: In section 3.2, the words of results should be slimed down because the main results in presented in Table 1.
Response 4: This is addressed as recommended. We reduced the narrative at the end of the last two sub-sections. Take a look at the revised version submitted.
Point 5: In Table 3, native or exotic species could be marked.
Response 5: The issue is addressed in the resubmitted version
Regarding comments on english minor revision.
Point 6: Line 80, few should be "a few"
Response 6: It is done as recommended.
Point 7: line 87, only sould be deleted.
Response 7: It's done.
Point 8: Line 93, "For Tang, Chen, & Zhao, (2016)", this kind of use should be in the style of citation. the authours used this kind of expression for figures, tables and citations some times.
Response 8: We corrected throughout the manuscript. This kind of expression has been replaced by "referring to the table, figure in the manuscript.
Thanks for taking time to make your observations which allow us to improve the manuscript.

Reviewer 3 Report
The authors collected a large and valuable scientific material: they organized 149 plots in the city, where they determined the taxonomic position of trees, measurest biomad their diameter and height, and calculated their mass. Ten species with highest biomass were listed. The authors are trying to determine the total mass of carbon stored in tree biomass within a city.
However, the work has many shortcomings in the statistical processing and presentation of data.
1. What is the main question addressed by the research?
- The question is to determine the carbon stocks in the biomass of trees in the territory of the city of Cotonou. This is important in connection with the problem of climate change. Differences in land use types are shown, which can help city management improve the situation.
2. Do you consider the topic original or relevant in the field? Does it
address a specific gap in the field?
-This article, to some extent, fills a gap in knowledge about the distribution of biomass.
3. What does it add to the subject area compared with other published
material?
-There is little data in the literature on the distribution of biomass in cities, and the article can make a significant contribution.
4.What specific improvements should the authors consider regarding the
methodology? What further controls should be considered?
- The authors use inadequate methods. For example, numbers 4.52 ± 5.24 (line 20), 3.00 ± 4.91, 2.39 ± 3.31 (table 3), etc. Parametric indicators are not suitable for describing the data. The authors write that the average number of individual trees is 13 per hectare; however, the authors studied 149 plots of 0.25 ha (37.25 ha) and 1536 trees, i.e., an average of 41 trees per hectare. Regression is applied for independent variables, but biomass depends on (Dbh)2 and height (diagrams in Fig. 3, page 8), so regression is not applicable. The diagram of the relationship between the diameter and height of trees (figure 3 on page 8) is questionable. The dots in the upper left corner of the diagram indicate that some trees with a minimum thickness of 10 cm had a maximum height of 24 m; this ratio is only possible for lianas. The number "7426 tons of carbon stored" is present only in the abstract on line 21; the text of the article does not explain how it was obtained; this number is not exact; probable limits are necessary. The authors need to either improve the reliability of these large data collections or improve them for greater clarity.
5. Are the references appropriate?
- References are rather appropriate. Chave et al. (2014) (line 149) is not included in the bibliography.
6. Please include any additional comments on the tables and figures.
- Figure 3 (page 6) shows the distribution of different diameter classes in terms of per hectare; the sum of the columns should be equal to the average number of individuals per hectare (according to the authors 13); however, in the diagram, the sum of the columns is >100. Diameter and height distribution diagrams are best presented for the entire sample rather than for the secondary quantity "individuals/ha".
The number of individuals studied (N) should be given in each row of tables 1,2, and 3.

Author Response
Dear Reviewer
Thank you for taking the time to provide quality comments on our manuscript despite your busy schedule.
Point 1: What is the main question addressed by the research?
The question is to determine the carbon stocks in the biomass of trees in the territory of the city of Cotonou. This is important in connection with the problem of climate change. Differences in land use types are shown, which can help city management improve the situation.
Response 1: We use this comment to improve the last part of the introduction of the manuscript. You can see it in the revised manuscript version.
Point 2: Do you consider the topic original or relevant in the field? Does it
address a specific gap in the field?
Response 2: The reviewer thinks that this article, to some extent, fills a gap in knowledge about the distribution of biomass.
Point 3: What does it add to the subject area compared with other published
material?
Response 3: The reviewer approved the adding value of the manuscript by confirming that there is little data in the literature on the distribution of biomass in cities, and the article can make a significant contribution.
Point 4: What specific improvements should the authors consider regarding the
methodology? What further controls should be considered?The authors use inadequate methods.
4.1: For example, numbers 4.52 ± 5.24 (line 20), 3.00 ± 4.91, 2.39 ± 3.31 (table 3), etc. Parametric indicators are not suitable for describing the data.
4.2: The authors write that the average number of individual trees is 13 per hectare; however, the authors studied 149 plots of 0.25 ha (37.25 ha) and 1536 trees, i.e., an average of 41 trees per hectare.
4.3: Regression is applied for independent variables, but biomass depends on (Dbh)2 and height (diagrams in Fig. 3, page 8), so regression is not applicable.
4.4: The diagram of the relationship between the diameter and height of trees (figure 3 on page 8) is questionable. The dots in the upper left corner of the diagram indicate that some trees with a minimum thickness of 10 cm had a maximum height of 24 m; this ratio is only possible for lianas.
4.5: The number "7426 tons of carbon stored" is present only in the abstract on line 21; the text of the article does not explain how it was obtained; this number is not exact; probable limits are necessary. The authors need to either improve the reliability of these large data collections or improve them for greater clarity.
Response 4:
4.1: On this point, we think that there may be a misunderstanding or the evaluator has not properly observed the table number we have used to describe the parametric indicators he is talking about. These parametric indicators are described from table 2 and not table 3 as he says. Table 3 describes only the top 10 species with a high carbon storage capacity.
4.2: In this respect, the reviewer's observations are well noted. It's true that we mentioned that 1,536 tree species had been counted, but not all the 1,536 woody species were used to calculate carbon stocks, as the original title of the manuscript indicates. In fact, the data collected as part of this research project has led to the production of three different manuscripts, the first two of which have already been published in MDPI's SUSTAINABILITY (june 2023) and CLIMATE (August 2023) journals respectively, in this year. The second, recently published on August 30 (two weeks ago), deals with plant diversity and species with high ecological value for urban resilience: the case of the city of Cotonou (https://www.mdpi.com/2225-1154/11/9/182). In this latest published article, the ecological data collected in the city concern floristic diversity (data for last published article) and also the dendrometric parameters (Dbh, Heights, which are for the current manuscipt) of tree species. These include woody and non-woody species. In the previous article on plant diversity in the city, we mentioned the number of plots (149) and the number of species in general (1536 plants), including woody and non-woody species. However, as this manuscript is concerned with assessing the carbon storage capacity of woody plants in the city of Cotonou, not all the 1536 plants mentioned are woody. Therefore, under your pertinent observation, this headcount will be deleted as all 1536 feet are not woody species used in the production of the manuscript.
Hoping this has convinced you, we have removed it from the manuscript.
4.3: In fact, the primary objective here is to show the evolution of the allometric relationship between dendrometric parameters (Height and Dbh) at the level of plant species in land-use units. To your concern, it's true that biomass depends on these two dendrometric parameters, but it depends more on tree diameter than on height. And this can be seen in the allometric equation used to calculate biomass. In this equation, diameter takes on more value and privilege than height. The diagram justifies the Chave et al. (2014) equation used to calculate biomass, and shows how these allometric relationships evolve across the city's different occupancy units. It may seem like a repetitionn with Table 1 but not all because it's the evolution and behavior that the figure exposes in the occupancy units that is more highlighted.
4.4: We think this statement does not apply to all trees. It's not only lianas that can meet this morphological tree characteristic. Let's take the example of Gmelina arborea, one of many species listed in our article published on August 30 in the journal climate MDPI (Special Issue Climate System Uncertainty and Biodiversity Conservation). This species (with 20 years old) can reach 30 m in height with a diameter of just 1.8 (even 2.4) cm (see MONOGRAPHIE GMELINA ARBOREA written by M. Boulet-Gercourt and published in Bois et forêts des Tropiques, n°172, March-April 1977).
4.5: You're quite right. But in fact, in the description of the study area, we said that the city of Cotonou covers an area of 79 kilometer . This was a calculation based on the surface area of the city in relation to the carbon rate estimated per hectare in the manuscript. But this did not actually appear until the date of the summary. This would have been explained in the result section, but we explained the estimated Carbon rate per hectare in the occupancy units. As it apears only in the summary, we prefere to cancele.
Point 5: Are the references appropriate?
- References are rather appropriate. Chave et al. (2014) (line 149) is not included in the bibliography.
Response 5: We addressed this issue as recommended. You can see it in the revised version of the manuscript.
Point 6.1 : Please include any additional comments on the tables and figures.
Response 6.1: The issue is addressed. For instance, we added this to figure 3 (a means position parameter (in cm), b is the average diameter and c is the shape parameter related to the diameter structure in question. According to Husch et al (2003), the Weibull can take several forms depending on the value of the shape parameter).
Point 6.2: Figure 3 (page 6) shows the distribution of different diameter classes in terms of per hectare; the sum of the columns should be equal to the average number of individuals per hectare (according to the authors 13); however, in the diagram, the sum of the columns is >100.
Response 6.2: In the figure, the distribution revealed is not in percent so that the sum gives 100. Nor have the authors 13 addressed questions of class distribution of dendrometric parameters of plant species.
Point 6.3: Diameter and height distribution diagrams are best presented for the entire sample rather than for the secondary quantity "individuals/ha".
Response 6.3: That's a nice proposal coming from you, but it depends on the objectives of the study which are also what we saw in the litterautrure and preview research on the study area. The objective here is to bring out the distribution by occupancy unit which gives a decentralized view of urban flora since the study was made with this objective in mind and policy recommendations will be made taking occupancy units into account. Better still, work within the city, for example Sehoun et al. (2020). Each piece of work is based on the objectives and recommendations that should be relevant to urban planning and development.
Point 6.4: The number of individuals studied (N) should be given in each row of tables 1,2, and 3.
Response 6.4: This logic is clearly visible in Table 2. The number of individuals per hectare in the land-use units (N, feet/ha) is indeed present in table 2. As for Table 1, this may not be present as it depends on the package (SMA here) used to carry out this statistical analysis. In fact, the table is the result of processing Standardized Major Axis regression (SMA) and the package does not take this into account, as it is already included in the ANOVA analysis of table 2. Table 3 shows the individual carbon stock capacities of the top 10 woody species.
Dear reviewer
we've tried to respond to all your comments. As a first step, we have made some changes to the manuscript where necessary. On the other hand, we have also explained scientifically where explanations were needed to justify our methodology and the results obtained.
Thank you for taking the time out of your busy schedule to read and make pertinent comments, which have helped us to improve the quality of this scientific document.
